# Effects of Sulfamethoxazole and Florfenicol on Growth, Antioxidant Capacity, Immune Responses and Intestinal Microbiota in Pacific White Shrimp *Litopenaeus vannamei* at Low Salinity

**DOI:** 10.3390/antibiotics12030575

**Published:** 2023-03-14

**Authors:** Yunsong Chen, Li Zhou, Qiuran Yu, Erchao Li, Jia Xie

**Affiliations:** 1Key Laboratory of Tropical Hydrobiology and Biotechnology of Hainan Province, Hainan Aquaculture Breeding Engineering Research Center, College of Marine Sciences, Hainan University, 58 Renmin Road, Haikou 570228, China; 2School of Life Sciences, East China Normal University, Shanghai 200241, China

**Keywords:** sulfamethoxazole, florfenicol, low salinity, growth and physiological performance, gut health, *Litopenaeus vannamei*

## Abstract

Antibiotic residue may pose a serious risk to aquaculture, and the culture of *Litopenaeus vannamei* in a low-salinity environment is a growing trend over the world. Here, we aimed to understand the combined effect of low salinity and sulfamethoxazole (SMZ) and florfenicol (FLO) antibiotics on *L. vannamei*. The growth performance, immune functions, antioxidant capacity and intestinal microbiota were investigated. Compared with the control group, the weight gain and survival rate significantly decreased (*p* < 0.05) in shrimp after they were exposed to low-salinity (salinity 3) water and the mixture of antibiotics and low-salt conditions for 28 days. The antioxidant activities of SOD and T-AOC, shown at low salinity and in the higher concentration of the SMZ treatment group (SMZH), were significantly decreased, while the GST activity was significantly increased in each treatment group in comparison with the control group. The expression of immune-related genes, including *TOLL*, *LvIMD*, *PPO* and *HSP*, in the low concentration of the SMZ treatment group (SMZL) was higher than that in the other groups. The diversity of intestine microbiota was disturbed with a lower Shannon index in the low-salinity and SMZH groups, and a higher Simpson index in the SMZH group. *Proteobacteria*, *Actinobacteria* and *Bacteroidetes* were the dominant phyla in the gut of *L. vannamei*. At the genus level, *Microbacterium*, *Shewanella*, *Aeromonas*, *Acinetobacter*, *Gemmobacter*, *Paracoccus* and *Lysobacter* were significantly decreased in the low-salinity group. However, the abundance of opportunistic pathogens belonging to the genus *Aeromonas* in the FLO group was increased. The predicted microbe-mediated functions showed that the pathway for “amino acid metabolism” and “replication and repair” was significantly inhibited in both the low-salinity and antibiotic-exposed groups. All the findings in this study indicate that the combined effect of antibiotics and low salinity on *L. vannamei* negatively impacted the physiological and intestinal microbiota functions.

## 1. Introduction

Antibiotics are widely used to prevent or treat bacterial infection in many fields, including human therapies, livestock, plant agriculture and aquaculture [1,2]. Due to their continuous release and partial degradation in the environment, antibiotics residue may cause ecological and human health risks [3]. A previous study reported that 92,700 t of antibiotics were used in China in 2013, 52% of which were consumed by animals [4]. Sulfonamide and chloramphenicols are commonly used in aquaculture [5]. Many studies have reported the detection of antibiotics in aquaculture in China [6,7,8], Asia [9,10], the USA [11] and European [12] countries. Previous studies found that antibiotics can induce oxidative stress [13], trigger immune and inflammatory responses [14] and change ecologic and metabolic functions of the intestinal microbiota of various aquatic species, including fish [15], crabs [16] and shrimps [17].

The Pacific white shrimp *Litopenaeus vannamei* is an important commercial species in aquaculture. It is bred worldwide and it has been adapted to a broad range of salinities (from 1 to 50 salinity units) and temperatures [18]. White shrimp have been cultured inland with low-salinity water (salinity < 5) in many countries due to economic needs or water pollution in some coastal areas [19]. Previous studies have indicated that white shrimp reared in low-salinity water showed slow growth, low survival and immunity, high sensitivity to pathogens and high-energy requirements [20]; in addition, this low-salinity environment disturbed the composition of their gut microbiota [21]. Joint environmental stress, such as changes in water conditions and antibiotic residues, exists in shrimp aquaculture. Earlier studies revealed the adverse effect of antibiotics or low salinity, though few studies have been conducted to investigate the combined stress of antibiotics and low salinity.

The study aims to investigate the potential chronic effect of antibiotics on shrimp in a low-salinity environment. The growth performance, antioxidant capacity, immune response and intestinal microbiota of *L. vannamei* were investigated after being exposed to experimental conditions for 28 days. This is the first study to explore the environmentally relevant concentrations of two fish antibiotics on the Pacific white shrimp under low-salt conditions.

## 2. Materials and Methods

### 2.1. Experimental Design

Sulfamethoxazole (SMZ ≥ 98%, Cas# 723-46-6) and florfenicol (FLO ≥ 98%, Cas# 73231-34-2) (China Shanghai Yien Chemical TechnCology Corporation, Shanghai, China) are commonly used in aquaculture. Juvenile *Litopenaeus vannamei* were purchased from a local farm in Hainan, China. After identifying the development stage with a microscope [22,23], the juvenile-stage shrimp were acclimated in aerated seawater for two weeks. The rearing water was collected from the nearshore of Haikou Bay and the antibiotic concentrations in the seawater were measured using HPLC before the experiment. Neither SMZ nor FLO were detected (detection limit is about 1 ng/L). The water parameters were temperature 27 ± 2 °C, pH 7.8–8.0, salinity 30%. During the acclimation period, the shrimp were fed three times daily (08:00, 12:00 and 18:00) with a commercial feed. The residual food and feces were removed with a siphon tube and 50% of the water from each tank was replaced daily.

After acclimation, 720 juveniles of *L. vannamei* (0.25 ± 0.012 g) were randomly divided into six groups with four replicates per treatment and 30 shrimps per tank. Based on the available literature on the environmental concentrations of the two antibiotics [6,24], the conditions were as follows: control group (salinity 30), 0 group (salinity 3), low concentration SMZ group (SMZL 6 ng/L, salinity 3), high-concentration SMZ group (SMZH 60 ng/L, salinity 3), low-concentration FLO group (FLOL 20 ng/L, salinity 3) and high-concentration FLO group (FLOH 200 ng/L, salinity 3). The size of the glass tank was 60 cm × 30 cm × 35 cm. All experimental shrimp were fed under the above conditions for 28 days, the residual waste (feces and food residual) was regularly processed every day and 50% of water was renewed with aerated water containing the selected doses of the antibiotics used.

### 2.2. Growth Evaluation and Sampling

The survival of shrimp in each tank was recorded every 24 h. After 28 days, all shrimp were deprived of food for 24 h. Then the body length, weight and number of shrimps in each tank were recorded. The parameters of growth performance were calculated according to our previous study [25].

### 2.3. Biochemical Assays

The hepatopancreas of two shrimp was taken from each tank, weighed and crushed with the prepared saline solution (0.86%, 1:10, *w/v*) using a multi-sample tissue grinder (Tissuelyser-24, Shanghai, China), then centrifuged (3–18 KS, Sigma Laborzentrifugen GmbH, Osterode, Germany) at 3000 rpm, 4 °C for 3 min. The supernatant was collected and used for biochemical assays. The total protein concentration, superoxide dismutase (SOD), catalase (CAT), total antioxidant capacity (T-AOC) and glutathione S-transferase (GST) in the shrimp hepatopancreas were detected using the colorimetric method with commercial assay kits (Nanjing Jiancheng Biological Co., Ltd., Nanjing, China).

### 2.4. Quantitative Real-Time PCR

The hepatopancreas of three shrimp was taken from each tank for total RNA extraction using TRIzol reagent (Invitrogen, Waltham, MA, USA). The quality and concentration of total RNA were measured using a NanoDrop 2000 spectrophotometer (Thermo, Wilmington, NC, USA). Then, real-time quantitative PCR analysis was performed on the total RNA (1 μg) using the HiScript II Q RT SuperMix for qPCR (+gDNA Wiper) kit (Vazyme Biotech, Nanjing, China) according to the manufacturer’s instructions. The expression of relative genes was performed by SYBR Green master mix (Vazyme Biotech, Nanjing, China) in a LightCycler© 96 system (Roche, Basel, Switzerland), and the internal reference gene was β-actin. Cycle time (Ct) of each treatment was compared to its corresponding internal control, and then converted to fold-change values by comparing it to the control group using the comparative 2^−ΔΔCT^ method. The primer sequences used in this study are shown in Table 1.

### 2.5. Intestinal Microbiota Analyses

A soil DNA kit (OMEGA, Georgia, GA, USA) was used to extract microbial DNA from intestine content samples according to the manufacturer’s instructions. The DNA concentration and quality were measured using a NanoDrop 2000 spectrophotometer (Thermo Scientific, Waltham, MA, USA). The forward primers 338F (5′ ACTCCTACGGGAGGCAGCAG 3′) and reverse primers 806R (5′ GGACTACHVGGGTWTCTAAT 3′) were used for amplifying the V3-V4 region of the bacterial 16S ribosomal RNA gene. The PCR reactions were conducted using the program reported in our previous study. Briefly, denaturation of 3 min at 95 °C, 27 cycles of 30 s at 95 °C, annealing of 30 s at 55 °C, elongation of 45 s at 72 °C and a final extension of 10 min at 72 °C. Triplicate PCR reactions were performed in a 20 μL mixture volume. According to standard protocols, purified amplicons were pooled in equimolar amounts and paired-end sequenced (2 × 300) on an Illumina MiSeq platform (Illumina, San Diego, CA, USA). Sequences have been submitted to the Sequence Read Archive (SRA) under the BioProject accession number PRJNA826220.

The raw sequence data were filtered using Q IIME. After quality control, Uparse software (Uparse v7.0.1090) was used for all sample clustering, with 97% consistency becoming operational taxonomic unit (OTU) sequence clustering; OTU representative sequences were selected at the same time. Species annotation was performed on OTU sequences, and species annotation analysis was carried out using the Mothur method and the SSSurrNA database of Silva138 (http://www.arb-silva.de/; accessed on 20 March 2022) (set thresholds of 0.8~1) (Edgar and Robert, 2013). RStudio (version 3.3.1) software was utilized to calculate the Venn diagram and to identify the shared and unique OTUs. Alpha diversity index, including Chao1, Shannon, ACE, and Simpson were calculated using Mothur software (version 1.30.2; http://www.mothur.org; accessed on 20 March 2022). To assess differences between microbial communities, we performed a principal coordinates analysis (PCoA) using Bray–Curtis distance metrics. Intestinal microbial functions were predicted using the bioinformatics package PICRUSt 2 (version 2.2.0), and the Kyoto Encyclopedia of Genes and Genomes (KEGG; http://www.genome.jp/kegg/; accessed on 20 March 2022) was used to annotate the predicted functional pathways at different levels.

### 2.6. Statistical Analyses

All data are expressed as the mean ± standard error (SEM). Statistical analysis was performed using SPSS statistics 25 (IBM, Armonk, NY, USA). One-way analysis of variance (ANOVA) and Duncan’s test were utilized to analyze the differences between experimental groups. The value of *p* < 0.05 was set for statistical significance.

## 3. Results

### 3.1. Growth Performance

Under a low-salinity environment, the weight gain and the survival of shrimp were significantly lower than the control (salinity 30) when exposed to SMZ and FLO (Figure 1A,B); however, the shrimp hepatosomatic index showed no differences between each group (Figure 1C). Compared to the 0 group (salinity 3), the condition factors of the shrimp in the high-concentration SMZ (60 ng/L, SMZH) and FLO (200 ng/L, FLOH) groups were significantly higher (Figure 1D).

### 3.2. Antioxidant Capacity and Immune Responses

Compared with the control group, the SOD activity of shrimp in the 0 group and SMZH groups was significantly lower (*p* < 0.05). For the shrimp in the SMZL and FLOH groups, SOD activity was significantly higher than that in the SMZH group (Figure 2A). No differences were found in the activities of CAT and T-AOC in shrimp from each treatment group when compared to the control group; however, the T-AOC activity of shrimp in the low-concentration SMZ (6 ng/L, SMZL) and FLO (20 ng/L, FLOL) group was significantly lower (*p* < 0.05) than in other treatment groups (Figure 2B,C). Moreover, the GST activity of shrimp exposed to salinity 3 water and SMZ or FLO was significantly higher than that in the control group (Figure 2D).

According to Figure 3, the relative expressions of immune genes including protein toll (*Toll*), immune deficiency (*LvIMD*), heat-shock protein 70 (*HSP70*) and prophenoloxidase (*pPO*) were significantly upregulated in the SMZL group when compared to the control group (*p*< 0.001). Moreover, when compared with 0 group, the *pPO* gene expression of shrimp was significantly increased in the FLOH group (Figure 3D).

### 3.3. Richness and Diversity of Intestinal Microbiota

A total of 737,371 high-quality sequences were obtained from the intestine microbiota using Illumina platform sequencing. A total of 201 OTUs were identified, among them, 69 OTUs were from the control group, 89 OTUs were from the 0 group (salinity 3), 64 OTUs were from the SMZL group, 110 were OTUs from the SMZH group, 79 were OTUs from the FLOL group and 90 OTUs were from the FLOH group. The number of unique OTUs in the control, 0, SMZL, SMZH, FLOL and FLOH groups were 9, 29, 4, 50, 19 and 30, respectively (Figure 4A). The comparison of microbiota compositions and the differences between microbiota were evaluated using beta diversity. Intestinal microbiota from the salinity 3 treatment shrimp was clearly separated from the control and SMZ or FLO treatment shrimp by PCoA analysis (Figure 4B).

The alpha diversity analysis at the genus level was calculated to evaluate the effects of salinity and antibiotic stress on the intestinal microbiota richness and diversity of *L. vannamei* (Figure 5). No differences were found in Chao1 and the Ace index between the control group and each treatment group (Figure 5A,C). However, compared with the control group, the Shannon index of shrimp intestinal microbiota in both the 0 and SMZH groups was significantly decreased (*p* < 0.05), while the Simpson index in SMZH group significantly increased (Figure 5B,D). Moreover, Chao1, Shannon, Ace and Simpson index in the SMZH group were significantly lower (*p* < 0.05) than in the FLOH group (Figure 5A–C).

### 3.4. Community Composition Analysis of Intestinal Microbiota

At the phylum level, *Proteobacteria*, *Actinobacteria* and *Bacteroidetes* were the dominant phyla in all groups (Figure 6A). Compared to the control group, *Bacteroidetes* abundances were significantly increased while *Cyanobacteria* abundances were significantly decreased in the 0 (salinity 3) group (*p* < 0.05). However, the proportions of *Verrucomicrobia* were significantly decreased in the SMZH and FLOL groups. At the genus level, *Microbacterium* was the dominant genus with the highest abundance in all groups (Figure 6B). Compared with the control group, *Microbacterium*, *Shewanella*, *Paracoccus*, *Lysobacter*, *Acinetobacter*, *Gemmobacter* and *Aeromonas* abundances were significantly decreased in the 0 group. Moreover, the abundance of *Lysobacter* was significantly increased in the SMZL group and *Pseudomonas* was significantly decreased in the SMZH group. In addition, *Aeromonas* was increased in the FLOL and FLOH groups.

### 3.5. Functional Predictions of Intestinal Microbiota

The intestinal microbiota function of *L. vannamei* based on PICRUSt was studied. In KEGG Level 1, a high abundance of bacterial metagenome in all the groups was associated with “Metabolism”, “Genetic information processing”, “Environmental information processing” and “Cellular processes”. In the level 2 term, “Amino acid metabolism”, “Carbohydrate metabolism”, “Metabolism of cofactors and vitamins”, “Metabolism of other amino acids” and “Cell motility” were decreased in the treatment groups when compared with those in the control group (Figure 7).

## 4. Discussion

A previous study has reported that the weight gain and survival of *L. vannamei* significantly decrease in low-salinity seawater; this is consistent with the research results in this experiment [27]. Antibiotics are used to promote biological growth, inhibit bacterial reproduction, prevent disease and improve immunity [28]. When compared with the low salinity groups (salinity 3), the weight gain, survival and hepatosomatic index of shrimp in each group showed no significant change after the addition of SMZ and FLO. It was reported that SMZ and FLO could inhibit the growth and development of zebrafish, microalgae and crustaceans [29]. The condition factors of shrimp in the SMZH and FLO groups were significantly higher than those in a single-low-salinity group, suggesting that antibiotics play a positive role in the general health of shrimp [30].

Reactive oxygen species (ROS) can help the body fight against pathogens, but excessive ROS will cause oxidative stress, resulting in oxidative damage, reduced biological immunity and inhibited growth and development [31]. Antioxidant enzymes such as SOD and CAT play an important role in removing excess ROS and maintaining homeostasis [32]. In this study, SOD activity was significantly decreased in the single-salinity 3 and SMZH groups when compared with the control. A previous study reported that 8 weeks of sulfamethoxazole exposure decreased the activity of antioxidant enzymes SOD in oriental river prawns [17]. However, low salinity induces an increase in SOD activity in *L. vannamei* after a 50-day trial [33]. These inconsistent results are possibly due to the exposure time and concentrations. T-AOC can stabilize the physiological level of the body and reflect the overall condition of white shrimp [34]. The SMZL and FLOL groups had lower T-AOC activities than the SMZH group and the FLOH group, which indicated that the white shrimp had different antioxidant states at different concentrations of SMZ and FLO. It has been reported that FLO could affect the activities of T-AOC in *L. vannamei* and have obvious time- and concentration-related effects [35]. GST is important in cellular detoxification. In this study, the GST activity of shrimp in low salinity and antibiotics treatment groups were significantly decreased; this is in agreement with the results of the previous study [36].

Research suggests that crustaceans once lacked a specific immune system; thus, they formed unique innate immunity during biological evolution to ensure their survival and reproduction [37]. Previous studies have shown that environmental stress can change the immune capacity of *L. vannamei* [38]. *Toll* and *LvIMD* are important non-specific immune genes that could help shrimps resist pathogens [39]. *Toll* expression is stimulated by Gram-positive bacteria, while *LvIMD* expression is enhanced by Gram-negative bacteria [40]. The expression of heat shock protein (HSP)-encoding genes is usually related to the stress of shrimp caused by temperature and pollution; these genes are involved in autoimmune diseases and innate immunity [41]. Moreover, the *proPO* system plays a crucial role in the innate immune responses in farmed species [38]. The current study found that the expression of *Toll*, *LvIMD*, *HSP70* and *pPO* in the SMZL group were significantly upregulated when compared with the control group. Similarly, immune-related genes were reported to be activated to cope with different stress on shrimps [39,42].

The structure of intestinal microbiota represents the health condition of shrimp. As the intestine of *L. vannamei* is not just involved in nutrient absorption but also related to immune capacity, the changes in the composition of the microbiota may cause diseases and increase mortality [43]. Previous studies have reported that environmental stress [44,45] and prebiotics [46,47] could induce a significant impact on the α-diversity of intestinal microbiota in shrimp. In this study, the alpha diversity of intestine microbiota was changed under the stress of low salinity and antibiotics. *Proteobacteria*, followed by *Actinobacteria* and *Bacteroidetes*, are the dominant phyla in *L. vannamei*; these could be regarded as stable bacteria in the intestine [48]. Compared to the control groups, *Bacteroidetes* had higher abundances in the low-salinity group while the proportions of *Cyanobacteria* were decreased. *Cyanobacteria* is associated with the production of cyclic lipopeptides and the decrease in *Cyanobacteria* may positively affect growth [49]. Moreover, *Verrucomicobia* can degrade polysaccharides and help digest feed fiber; as such, it is a potential glycoside hydrolysis generalist [50]. In this study, the abundance of *Verrucomicbia* decreased in the SMZH and FLOH groups, which suggests that the addition of antibiotics in low-salinity conditions can interrupt the digestion function of *L. vannamei* gut. The decrease of “*Microbacterium*” has been reported to slow growth for aquatic animals and “*Shewanella*” were reported significantly decreased in *L. vannamei* under the stress of sulfide [51]. “*Shewanella*”, “*Aeromonas*” and “*Acinetobacter*” are opportunistic pathogen bacteria of shrimp, while “*Lysobacter*” has an antagonistic effect on pathogens [52]. “*Rhodobacteraceae*” is a probiotic that can promote vitamin synthesis by symbiotic bacteria to ensure homeostasis in a low-salt environment [53]; “*Gemmonbacter*” belongs to this family [54]. In the present study, the abundance of “*Shewanella*”, “*Rhodobacteraceae*”, “*Lysobacter*”, “*Acinetobacter*”, “*Gemmonbacter*” and “*Aeromonas*” was significantly changed in the low-salinity groups when compared to the control group. The potential probiotics “*Rhodobacteraceae*” and “*Pseudomonas*” were decreased in the SMZH group, and the opportunistic pathogens “*Aeromonas*” increased in the FLO groups. This suggests that the addition of SMZ may weaken the intestinal barrier function of shrimp.

In this study, PICRUSt function prediction analysis revealed that the metabolism, genetic information processing, environmental information processing and cellular processes between the control and each treatment group were significantly different. Amino acid metabolism was reported to play an important role in the osmoregulation of *L. vannamei* in a low-salinity environment [55]. The metabolism of “tryptophan”, “starch and sucrose”, “nicotinate and nicotinamide”, “arginine biosynthesis” and “glutathione” was significantly decreased in SMZ groups. A previous study reported that amino acids were abundantly consumed as an energy source when animals were exposed to low salinity [56]. In this regard, we speculate that the addition of SMZ may reduce the ability of *L. vannamei* in amino acid-based osmoregulatory systems of low-salinity adaptation. In addition, the “replication and repair” pathway was increased significantly in *L. vannamei* under the stress of WSSV infection [57], and temperature fluctuations [58] were significantly decreased by SMZ and FLOL in the present study. This inconsistency may be due to the co-effect of low salinity and antibiotics.

## 5. Conclusions

In this study, shrimp in both low-salinity and antibiotic-addition conditions showed lower growth and survival rates. The antioxidant capacity of shrimp after exposure to low salinity and 60 ng/L SMZ was decreased. However, the expression of immune-related genes in shrimps was significantly upregulated by SMZ. Chronic low salinity and antibiotic co-exposure changed the diversity of gut microbiota and increased the opportunistic pathogens. Overall, the use of antibiotics in a low-salinity culture may cause a more negative effect on *L. vannamei*.

## Figures and Tables

**Figure 1 antibiotics-12-00575-f001:**
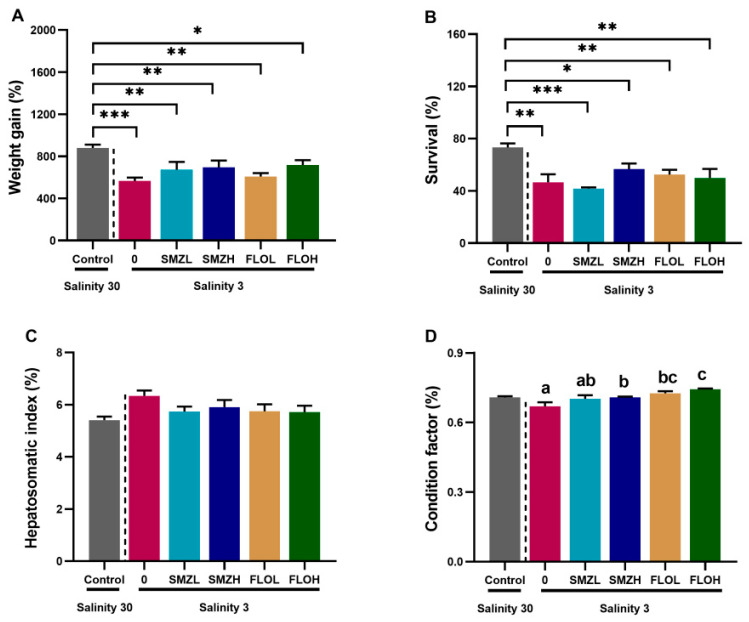
Weight gain (**A**), survival (**B**), hepatosomatic index (**C**) and conditional factor (**D**) of *L. vannamei* exposed to SMZ and FLO under low salinity environment for 28 days. All data are expressed as the mean ± SE (n = 4). Different letters (a–c) and * *p* < 0.05, ** *p* < 0.01, *** *p* < 0.001 indicate significant differences among groups.

**Figure 2 antibiotics-12-00575-f002:**
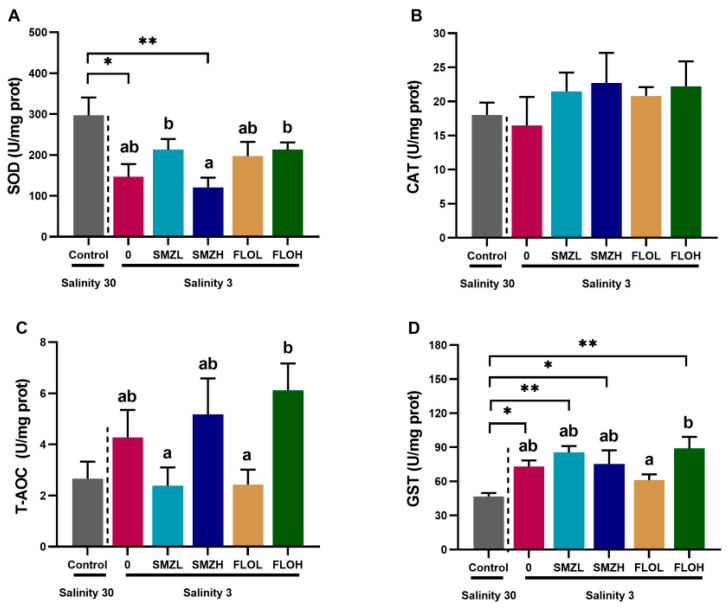
Hepatopancreas oxidative stress response of *L. vannamei* exposed to SMZ and FLO under low salinity environment for 28 days. SOD activity (**A**), CAT activity (**B**), T-AOC activity (**C**) and GST activity (**D**). All data are expressed as the mean ± SE (n = 4). Different letters (a,b) and * *p* < 0.05, ** *p* < 0.01 indicate significant differences among groups.

**Figure 3 antibiotics-12-00575-f003:**
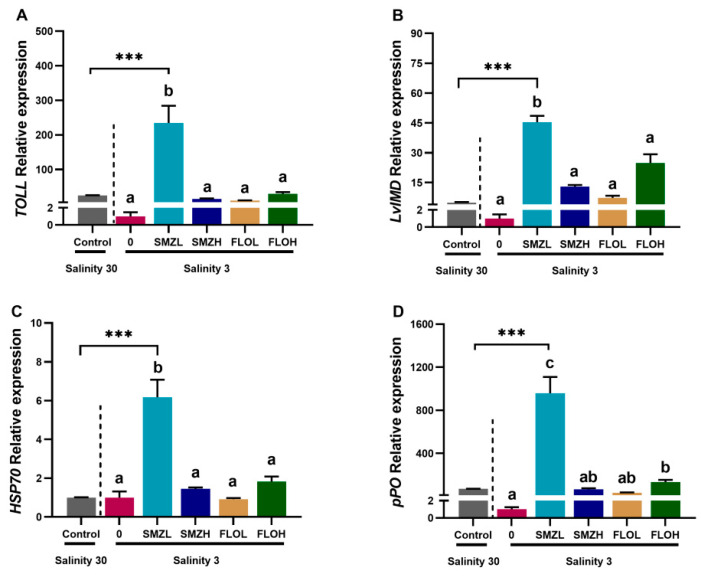
Innate immune responses of *L. vannamei* exposed to SMZ and FLO under low salinity environment for 28 days. *TOLL* (**A**), *LvIMD* (**B**), *HSP70* (**C**) and *PPO* (**D**) mRNA expression. All data are expressed as the mean ± SE (n = 4). Different letters (a–c) and *** *p* < 0.001 indicate significant differences among groups.

**Figure 4 antibiotics-12-00575-f004:**
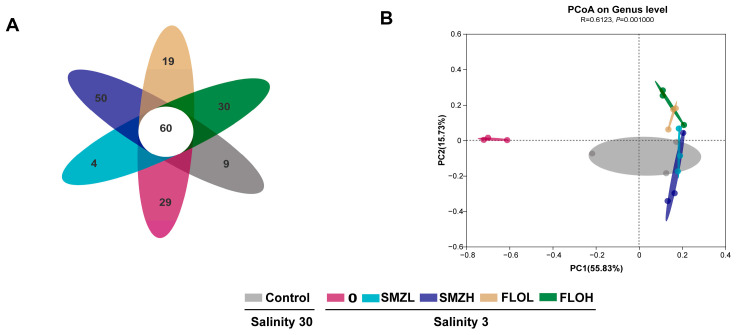
Differences between intestine microbiota communities of *L. vannamei* exposed to SMZ and FLO under low salinity environment for 28 days. Venn diagram (**A**), PCoA analysis (**B**).

**Figure 5 antibiotics-12-00575-f005:**
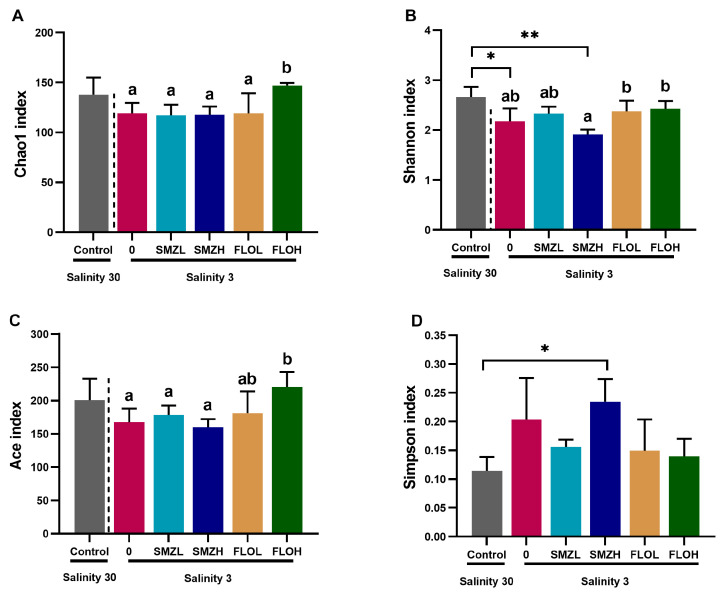
Alpha diversity indices of intestinal microbiota in *L. vannamei* exposed to SMZ and FLO under low salinity environment for 28 days. Chao1 index (**A**), Shannon index (**B**), Ace index (**C**) and Simpson index (**D**). All data are expressed as the mean ± SE (n = 3). Different letters (a,b) and * *p* < 0.05, ** *p* < 0.01, indicate significant differences among groups.

**Figure 6 antibiotics-12-00575-f006:**
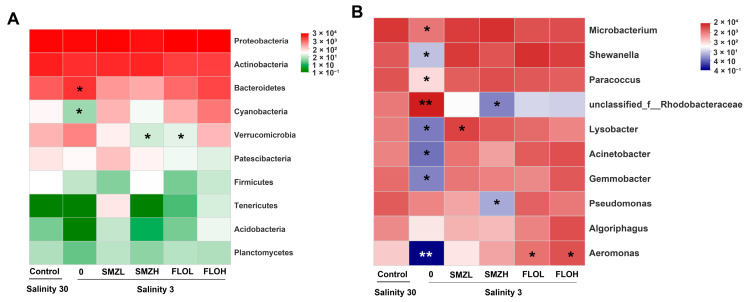
Composition at the phylum level (**A**), composition at genus level (**B**), * *p* < 0.05 and ** *p* < 0.01 indicates a significant difference.

**Figure 7 antibiotics-12-00575-f007:**
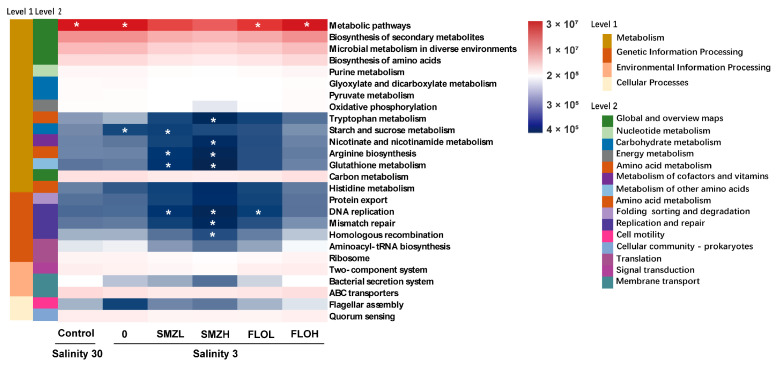
KEGG pathways analysis by PICRUSt, * *p* < 0.05 indicates a significant difference.

**Table 1 antibiotics-12-00575-t001:** Primer sequences used in this study.

Primer Name	Sequence (5′-3′)	GenBank Accession Number
*Toll-F*	GACCATCCCTTTTACACCAGACT	DQ923424
*Toll-R*	CCTCGCACATCCAGGACTTTTA	
*LvIMD-F*	TGGGTCCGTGTCCAGTGAT	[26]
*LvIMD-R*	ACAAACAACCACACACAAGCAG	
*HSP70-F*	CAACGATTCTCAGCGTCAGG	XM_027369405
*HSP70-R*	ACCTTCTTGTCGAGGCCGTA	
*pPO-F*	CAATGACCAGCAGCGTCTTC	AY723296
*pPO-R*	CACGGAAGGAGGCGTATCAT	
*β-actin-F*	GCAGTCCAACCCGAGAGGAAG	AF300705
*β-actin-R*	GTGCATCGTCACCAGCGAA	

## Data Availability

The data provided in this study have been uploaded to the NCBI database. The accession number is PRJNA826220, and the links are https://www.ncbi.nlm.nih.gov/sra/PRJNA826220 (accessed on 8 June 2022).

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
