# Peer review of "Effects of Sulfamethoxazole and Florfenicol on Growth, Antioxidant Capacity, Immune Responses and Intestinal Microbiota in Pacific White Shrimp Litopenaeus vannamei at Low Salinity"

_antibiotics, 2023, doi:10.3390/antibiotics12030575_

Round 1

Reviewer 1 Report

Regarding the manuscript, Some clarification of the number of samples taken from each tank for the intestinal microbiota analysis, also there is discripanancy between sample number in different test for example for the biochemical assay and Real time PCR, the samples taken for theses tests are not equal and very low which is not representing the whole group number. my concern is the conclusion drown underestimate the group due to low number of samples.

English improvement 

Author Response

Point 1: Regarding the manuscript, Some clarification of the number of samples taken from each tank for the intestinal microbiota analysis, also there is discripanancy between sample number in different test for example for the biochemical assay and Real time PCR, the samples taken for theses tests are not equal and very low which is not representing the whole group number. my concern is the conclusion drown underestimate the group due to low number of samples.

English improvement .

Response 1: We thank the reviewer for this comment. The number of samples for the biochemical assay, Real time PCR and intestinal microbiota analysis were correspond to biological replicates.

Our manuscript has improved by a English-editing service. Please see the approved file.

Reviewer 2 Report

Antibiotic residue may pose a serious risk to aquaculture, and the culture of Litopenaeus vannamei in a low salinity environment is a growing trend over the world. Here, we aim to understand the combined effect of low salinity and antibiotics, Sulfamethoxazole (SMZ) and florfenicol (FLO) on L. vannamei. Growth performance, immune functions, antioxidant capacity and intestinal microbiota were investigated.

This manuscript is interesting and could be considered for publication.

Comments:

1. The introduction also displays data from various countries, not just China.

2. How is "Juvenile shrimp Litopenaeus vannamei" identified and authenticated? by whom? which laboratory? it should be given details on methods.

3. All abbreviations must be accompanied by a definition at the beginning of their presentation. For example, at the beginning of the Results section "SMZ (SMZH) and FLO (FLOH)" what does this stand for?

Check all abbreviations in the manuscript.

Author Response

Antibiotic residue may pose a serious risk to aquaculture, and the culture of Litopenaeus vannamei in a low salinity environment is a growing trend over the world. Here, we aim to understand the combined effect of low salinity and antibiotics, Sulfamethoxazole (SMZ) and florfenicol (FLO) on L. vannamei. Growth performance, immune functions, antioxidant capacity and intestinal microbiota were investigated.

This manuscript is interesting and could be considered for publication. 

Point 1:  The introduction also displays data from various countries, not just China.

Response 1: We thank the reviewer for this comment. We have added some references about the level of antibiotics from other countries, and marked it in red color. Please see page 2, lines 43-44.

Point 2: How is "Juvenile shrimp Litopenaeus vannamei" identified and authenticated? by whom? which laboratory? it should be given details on methods.

Response 2: We thank the reviewer for this comment. “Juvenile shrimp Litopenaeus vannamei” was identified by observing with microscope following the methods reported by Wei et al. (2014) and  Hertzler and Freas et al. (2009). We have add the references in the manuscript, and marked it in red color. Please see page 2, line 71.

Point 3: All abbreviations must be accompanied by a definition at the beginning of their presentation. For example, at the beginning of the Results section "SMZ (SMZH) and FLO (FLOH)" what does this stand for? Check all abbreviations in the manuscript.

Response 3: Thanks for the reviewer’s comment. We have revised in the manuscript, and marked it in red color. Please see pages 4-5, lines 158, 170-171.  

Reviewer 3 Report

The experimental design when the antibiotic dosages were not suitable for comparisons, this did not discredit the publication, but a design in a factorial scheme or even increasing levels separately for each product would be the most recommended.

The conclusion should be limited to the last sentence.

Author Response

Point 1: The experimental design when the antibiotic dosages were not suitable for comparisons, this did not discredit the publication, but a design in a factorial scheme or even increasing levels separately for each product would be the most recommended.

The conclusion should be limited to the last sentence.

Response 1: Thanks for the reviewer’s comment. The selection of the dose of antibiotics was on the available literature (Chen et al., 2015; Du et al., 2017) on the environmental concentrations of the two antibiotics. We have added relevant references in the manuscript, and marked it in red color. Please see page 2, lines 80-81.

We have revised the last sentence of conclusion: “Overall, the use of antibiotics in low salinity culture would cause more negative effect on L. vannamei” was replaced by “Overall, the use of antibiotics in low salinity culture may cause more negative effect on L. vannamei”. Please see  page 10, line 330.

Reviewer 4 Report

This manuscript provides some interesting and novel data regarding the impacts of co-exposure to low salinity and antibiotic on some physiological parameters and gut composition in Litopenaeus vannamei. However, the data obtained in this study were not well discussed and the English writing of this manuscript is not acceptable as there were a lot of grammatical errors within the text. This manuscript cannot be accepted in the present form and major corrections are required as follows:

Lines 2-3. Please change this part of the manuscript title “antioxidant, immune systems”.

Line 21. Please rewrite “in the lower addition of SMZ group (SMZL) than in the other groups”.

Line 26. Please change “opportunistic pathogens Aeromonas” to “opportunistic pathogens belonging to the genus Aeromonas”.

Line 29. Please change “addition” to “exposed”.

Line 38. Please change “partially degraded” to “partial degradation”.

Line 43. Should be “trigger”.

Line 47. Please change “has been bred worldwide and adapted” to “is bred worldwide and has been adapted”.

Line 53. Please remove “and construction”.

Lines53-54. Please rewrite “Joint environmental stress, such as changes in water condition, antibiotics residues are existed in shrimp aquaculture”.

Line 55. Please change “are revealed” to “revealed”.

Line 60. Please modify “L. vannamei were investigated after exposed” to “were investigated after exposure of L. vannamei”.

Line 69. Please specify the source of seawater used.

Line 79. What was the basis for the selection of doses used in this study? Did the authors test the LC50 for the antibiotics applied in this study? Please add a relevant reference.

Line 82. Please add a sentence here that 50% of water was renewed with aerated water containing the selected doses of the antibiotics used.

Line 101. Please rewrite “Then, real-time quantitative PCR analysis was performed of total RNA (1 μg) by using”.

Line 105. Please elaborate on the method used for normalization of the gene expression data and the group was used for normalization.

Line 106. Please change “show as” to “are shown”.

Line107. In table 1, the accession number used for heat shock protein should be changed (AY645906.1 to XM_027369405). Please also remove “.1” from all accession numbers.

Line 121. Please change “Sequences have been submitted to GenBank with serial numbers PRJNA826220” to “Sequences have been submitted in the Sequence Read Archive (SRA) under the BioProject accession number PRJNA826220”.

Lines 157-158. Please rewrite “and the shrimp in SMZL and FLOH groups were significantly higher than that in SMZH group”.

Line 187. Please change “after” to “in”.

Lines 218, 220 and 221. “Aeromonas” and “Pseudomonas” should be italic.

Line 246. Should be “important”. Please change “protecting” to “maintaining”.

Line 250. Please remove “was reported”.

Line 251. Please modify “results possibly” to “results are possibly”.

Line 261. Please rewrite “Research suggest that crustaceans were lack of”.

Line 265-266. Please change “Toll expression stimulates gram-positive bacteria and LvIMD expression stimulates gram-negative bacteria” to “Toll expression is stimulated by Gram-positive bacteria, while LvIMD expression is enhanced by Gram-negative bacteria”.

Line 277. Please remove “been”.

Line 280. Should be “followed”.

Line 282. Should be “bacteria”.

Line 285. Please revise “may good to growth” and provide a scientific reason.

Line 291. Please change “What more” to “Moreover”.

Line 299. “Aeromonas” should be italic.

Line 300. Please change “weak” to “weaken”.

Line 305. Should be “to play”.

Line 308. Should be “decreased”. Please add a relevant reference in association with your speculation!?

Lines 312-313. Should be “inconsistency”.

Line 316. Should be “exposure”.

Line 320. Please change “would cause” to “may cause”.

Round 2

Reviewer 4 Report

The authors addressed all the revisions requested and I suggest accepting this manuscript in the present form.